# Identifying Potentially Climatic Suitability Areas for *Arma custos* (Hemiptera: Pentatomidae) in China under Climate Change

**DOI:** 10.3390/insects11100674

**Published:** 2020-10-04

**Authors:** Shiyu Fan, Chao Chen, Qing Zhao, Jiufeng Wei, Hufang Zhang

**Affiliations:** 1College of Plant Protection, Shanxi Agricultural University, Jinzhong 030801, China; victoriafan@stu.sxau.edu.cn (S.F.); chenchao01@tyut.edu.cn (C.C.); zhaoqing@sxau.edu.cn (Q.Z.); wjfeng@nwsuaf.edu.cn (J.W.); 2Department of Biology, Xinzhou Teachers University, Xinzhou 034000, China

**Keywords:** *Arma custos*, MaxEnt, climate change, climatic suitability, ecological niche model, *Spodoptera frugiperda*

## Abstract

**Simple Summary:**

*Arma custos* is a predatory insect that can attack *Spodoptera frugiperda* and many other important agricultural and forest pests. In this study, we built a model to predict the potential distribution of *A. custos* under current and future climatic conditions for supporting its current and future use. Currently, the potential highly suitable areas of *A. custos* are mainly distributed in Hebei, Henan, Shandong, Anhui, Hubei, Jiangsu, and Zhejiang Provinces. Under the climate change scenarios of RCP2.6, 4.5, 8.5 in the 2050s and 2070s, the suitable areas for *A. custos* will decrease and shift towards Northeast China. Considering the currently suitable distribution area of *S. frugiperda*, artificially reared *A. custos* is suitable for release in Fujian, Zhejiang, Jiangxi, Hunan, and southeastern Sichuan Provinces under the current climatic condition. Under the future climate scenarios, Northeast China is not suitable for the survival of *S. frugiperda.* Thus, *A. custos* does not need to be released here.

**Abstract:**

*Spodoptera frugiperda* is a notorious pest that feeds on more than 80 crops, and has spread over 100 countries. Many biological agents have been employed to regulate it, such as *Arma custos*. *A. custos* is a polyphagous predatory heteropteran, which can effectively suppress several agricultural and forest pests. Thus, in order to understand where *A. custos* can survive and where can be released, MaxEnt was used to predict the potentially suitable areas for *A. custos* in China under climate change conditions. The results show that the annual mean temperature (bio1) and annual precipitation (bio12) are the major factors influencing the distribution of *A. custos.* The optimal range of the two are 7.5 to 15 °C, 750 to 1200 mm, respectively. The current climate is highly suitable for *A. custos* in Hebei, Henan, Shandong, Anhui, Hubei, Jiangsu, and Zhejiang Provinces. Considering the currently suitable distribution area of *S. frugiperda*, artificially reared *A. custos* is suitable for release in Fujian, Zhejiang, Jiangxi, Hunan, and southeastern Sichuan Provinces. Under the future climatic scenarios, the suitable area will decrease and shift towards the north. Overall, this result can provide a reference framework for future application of *A. custos* for biological control.

## 1. Introduction

The global transportation and trade have caused the spread of invasive species, greatly threatening crop production globally [1]. This constitutes a serious threat to developing countries that have poor agricultural productivity and huge food demands. Among the invasive pests of economic importance, the fall armyworm (*Spodoptera frugiperda*, FAW) is one of the most aggressive pests due to its strong flight ability and adaptation to a wide range of environments [1,2,3]. FAW is a polyphagous pest with broad host range, known to feed on 353 host plants globally, especially maize [4,5]. FAW is native to North America but has spread rapidly to at least 40 countries of sub-Saharan Africa [6]. In recent years, FAW has been detected in several Asian countries, such as India, Sri Lanka, Thailand, Yemen, Myanmar, and China [7,8,9]. In China, FAW was first discovered in Yunnan Province in late December 2018. Until now, it has spread to 26 provincial regions of the whole country and the damaged area exceeds one million ha, constituting a severe threat to food security in China [8,9,10].

After a severe worldwide outbreak of FAW, chemical insecticides were used as the main control method [11]. However, due to the intensive application of pesticides, FAW has developed resistance in several countries [12,13]. In order to avoid insecticidal resistance and protect environment, it is better to apply biological control to regulate FAW in the long term. There are many effective biological measures against FAW, such as natural enemies [14] and pathogenic microorganisms (fungi, bacteria, viruses, and nematode) [15,16,17,18,19]. The predators of FAW mainly include Coccinellidae, Carabidae, Reduviidae, Lygaeidae, and Pentatomidae [20]. In America and Brazil, *Podisus maculiventris* and *Podisus nigrispinus* have been applied to control FAW [21,22]. In China, many parasitoids, such as *Microplitis simili* [23] and *Chelonus munakatae* [24], as well as some predatory insects, such as *Arma custos* [25], have been used to check the continuous spreading of FAW.

*Arma custos* (Hemiptera: Pentatomidae) is an important predaceous insect, which is adaptable and easy to mass produce [26]. It is widely distributed in the Palaearctic and frequently found on various trees, as well as in cotton and soybean fields [27,28]. *A. custos* has received global attention because of its ability to effectively suppress several pests, including species of Lepidoptera, Coleoptera, Hymenoptera, and Hemiptera [28,29,30,31]. *A. custos* is suitable for artificial rearing and released adults easily establish a natural population under suitable ecological conditions, providing continuous control of the pest population [32]. In China, *A. custos* has been successfully applied to control many important pests. As early as the 1970s, some researchers have released artificially reared *A. custos* to control *Ambrostoma quadriimopressum* and obtained good effects [33]. Since then, some other institutes have applied *A. custos* to test pests, such as *Parocneria furva*, *Stilpnotia candida*, *Cnidocampa flavescens,* and achieved good control effects [34]. *A. custos* could also prey on FAW according to laboratory and field experiments. *A. custos* is most likely to attack FAW when in its 4th–5th nymphal instars, attacking mostly 3rd instar FAW larvae [35]. Among all instars of *A. custos*, the 5th instars are more active in predation of FAW [36]. At present, most of the researches about *A. custos* focus on its biological characteristic [28], morphology [37], artificial rearing [38], storage technology [39], and the predatory ability [40]. However, it is essential to know the potential distribution of *A. custos* for better sampling and releasing to control FAW and other pests.

Ecological Niche Models (ENMs) are widely used in ecology, biogeography, conservation biology and other fields, which can predict species potential distribution based on its known occurrence records and environmental predictors [41,42]. Among the ENMs, MaxEnt is one of the most widely used models by finding the probability distribution of maximum entropy. It is based on presence-only data and has high stability with small sample size [43,44,45]. Thus, in this study, MaxEnt was used to identify potentially suitable areas for *A. custos*.

This study attempts to apply *A. custos* to regulate FAW and also provide a theoretical reference for the control of other pests by predicting potentially suitable areas of *A. custos*. Overall, the objectives are: (i) to identify the most important environmental variables that influence the distribution of *A. custos*; (ii) to study the effect of climate change on the potential distribution of *A. custos*; and (iii) to provide a theoretical basis for applying *A. custos* to biological control.

## 2. Materials and Methods

### 2.1. Occurrence Data

The occurrence data of *A. custos* were primarily obtained from two sources: (i) the Global Biodiversity Information Facility database (GBIF, https://www.gbif.org/); and (ii) specimens by field collection. We extracted geographical coordinates of distribution sites based on Google Earth and some data were collected from the field by using global positioning system (GPS) (Appendix A-1, Appendix A-2). 

We first performed initial cleaning of occurrence data. The distribution sites without coordinates, low precise (decimal < 2), and duplicate coordinates were removed from initial data. Sample bias is frequently present in the occurrence records because the selection of sampling sites is easy to be subjectively selected as accessible areas, such as areas close to cities or other human settlements [46,47]. In order to eliminate imbalance and spatial autocorrelation caused by sampling bias, occurrence records were sub-sampled based on the function “gridSample” of R package “dismo”. After cleaning and filtering, 271 occurrence points remained (Figure 1). The workflow was conducted in QGIS version 3.12.2 (https://www.qgis.org/).

### 2.2. Environmental Variables

Climatic and topographic variables are used to characterize species niches in multivariate environmental space [48]. In a large spatial scale, climatic variables are considered as the primary factors to determine species niches [49,50]. In this study, 20 environmental variables including 19 bioclimatic variables (bio1-bio19) and one elevation data (altitude) for both current and future scenarios (Appendix A) were downloaded from the WorldClim database (https://www.worldclim.org/) with 2.5 arc-minute spatial resolution (about 5 km at the equator). Two topographic factors, i.e., slope (slo) and aspect (asp), were extracted from altitude in QGIS version 3.12.2. 

Data for future climatic conditions (2050s, average for 2041–2060; 2070s, average for 2061–2080) were derived from downscaled global circle models (GCMs). The Beijing Climate Center System Model version 1.1 (BCC-CSM1-1) is one of the most commonly used models for climate change simulation in China [51,52]. Thus, the present study adopted the GCM to predict the future suitable distribution of *A. custos*. In the fifth report assessment (AR5) of the International Panel on Climate Change (IPCC), four representative concentration pathways (RCPs) (RCP2.6, RCP4.5, RCP6.0, and RCP8.5) were established, which represented the possible future emission of greenhouse gases [53,54]. The four RCPs, ranging from lowest (RCP2.6) to highest (RCP8.5) values, are defined by the possible range of global radiative forcing values (2.6, 4.5, 6.0, and 8.5 W/m^2^, respectively) in the year 2100. To better understand the change of suitable distribution of *A. custos* under different levels of climatic scenarios, RCP2.6 as the minimum emission scenario, RCP4.5 as the medium, RCP8.5 as the maximum, were selected to simulate the potential future distribution in the 2050s and 2070s.

Variable selection is a crucial step for species distribution modeling. Both large and small subsets of environmental layers can greatly impact the performance of the models [55]. Model built with high multi-collinear variables is easily over-fitting [56]. Hence, in the current study, three methods (principal components analysis, Pearson correlation analysis, and the jackknife analysis) were combined to select the environmental variables with low correlation but high significance. Principal components analysis (PCA) and Pearson correlation analysis, as the main correlation-reducing techniques, are both widely used in the process of variable selection [57,58]. First, five principal components (PC1-PC5) were selected by PCA performed in R package “kuenm”, which explained almost 86% of variation in environmental variables. The three variables with highest correlation coefficient were selected from the five principal components as principal variables (Appendix A-1, Appendix A-2). Next, correlation analysis was performed using the R package “ellipsenm” and retained only one variable from each pair of highly correlated variables (│r│ ≥ 0.8), based on the percent contribution of each variable in the initial model by jackknife analysis in MaxEnt (Figure 2, Appendix A and Appendix A). Then, combining the result of PCA, six variables with low correlation but high importance (bio1, bio12, bio15, bio16, bio17, alt) were screened. 

### 2.3. Modeling Procedure

There are many ENMs used for modeling prediction, such as CLIMEX, BIOCLIM, and GARP [45,59,60]. MaxEnt is a presence-only model that performs well regardless of the number or geographical extent of species records as compared to the performance of other methods [43,50,61]. In this study, MaxEnt version 3.4.1 was used.

Species distribution parameters in MaxEnt determine the performance of the models. However, many pieces of researches adopted default parameters to execute models, which could result in severe fitting deviation for models [48]. Regular Multiplier (RM) and Feature Class (FC) are included in MaxEnt for optimizing the models. RM promotes to smooth the model and minimize model over-fitting. FC, corresponding to the response type of suitability values to each variable, determines the potential shape of response curves. There are five FCs in MaxEnt, which include Linear (L), Product (P), Quadratic (Q), Threshold (T), and Hinge (H) [43,62]. In the process of parameter optimization, the RM value ranged from 0.5 to 4 with an increment of 0.5 and six combinations of FCs (L, LQ, LQP, LQT, QPT, PHT), were chosen to build candidate models for selecting the optimized model. The R package “ENMeval” was used to execute the process and “the checkerboard2” method of “ENMeval” was applied to calculate the standardized Akaike information criterion coefficient (AICc), which has a criterion for evaluating the models. The lowest AICc score with the sets of parameters was selected to run the final model in MaxEnt. In the study, the optimal parameter set was PHT for FC and 1.5 for RM (Appendix A, Appendix A).

In the process of MaxEnt outputting, the convergence threshold and the maximum number of iterations were set to 10^−5^ and 500, respectively. Ten-fold cross-validation was used to execute MaxEnt for preventing random errors from the predicted samples, which were randomly partitioned into ten equivalent subsets, in which one was for model testing, and nine for model training. The logistic format was set to illustrate the results of MaxEnt. The binary suitable/non-suitable habitats were defined by a 10th percentile training presence logistic threshold, which has been widely applied to species distribution modeling, especially when data were collected by different collectors [63]. We adopted the median of all the bootstrapped replicates to better present the geographic predictions of final models [64]. The distribution maps were classified into four levels: <threshold, unsuitable; threshold–0.4, marginally suitable; 0.4–0.6, moderately suitable; and 0.6–1, highly suitable.

### 2.4. Model Evaluation

There are many indices applied to model evaluation, such as the area under the curve of the receiver operating characteristic (AUC), kappa statistic (kappa), and the true skill statistic (TSS) [65]. Although AUC is widely used, it can neither provide information on the spatial distribution of model errors nor weigh omission and commission errors equally [50,66,67]. To handle this problem, the partial AUC (pAUC) was applied to assess model performance. This metric gives priority to omission error over commission error and could consider the amount of error known or estimated among occurrences. In addition, it was recommended to set the type of replication as bootstrap running 1000 replicates [68]. In the study, pAUC was calculated by NicheToolbox (http://shiny.conabio.gob.mx:3838/nichetoolb2/) with 1000 iterations and error (E) = 0.05.

## 3. Results

### 3.1. Model Performance

In the present study, the mean value for pAUC (at E = 0.05) was 0.811777 (mean AUC: 0.891), which represented good credibility for the model. Moreover, the distribution of the AUC ratio calculated as AUC_partial_/AUC_random_ was significantly greater than the random AUC ratio (*p* < 0.001), which shows high performance of the model (Figure 3).

### 3.2. Effects of Environmental Variables

#### 3.2.1. Contributions of Environmental Variables

The annual mean temperature (bio1) and the annual precipitation (bio12) were the major factors influencing the distribution of *A. custos*. The contributions of bio1 and bio12 were 58.4% and 24.8%, respectively, which contained most of the environmental information (Table 1). Other variable contributions were: precipitation seasonality (bio15), 1.9%; precipitation of wettest quarter (bio16), 4%; precipitation of driest quarter (bio17), 8.6%; and altitude (alt), 2.3%.

#### 3.2.2. Response to the Environmental Variables

The response curves show how the climatic suitability of *A. custos* changes with the environmental variables (Figure 4). The suitability of *A. custos* decreased after the altitude (alt) reached about 150 m, where the curve trends for alt and precipitation seasonality (bio15) were roughly equal. Before the annual mean temperature (bio1) reached about 8 °C, there was a rise in the curve with the increasing temperature, and the rate of increase gradually decreased after 15 °C. The curve trend of bio1 was roughly consistent with that of the precipitation of driest quarter (bio17). When the annual precipitation (bio12) reached between 1200 mm to 1400 mm, the suitability of *A. custos* dropped sharply and the curve of bio12 varied consistently with the precipitation of wettest quarter (bio16). Overall, the response curves illustrated that the high probability of presence of *A. custos* was at altitude (alt) of 0 to 150 m, annual mean temperature (bio1) of 7.5 to 15 °C, annual precipitation (bio12) of 750 to 1200 mm, precipitation seasonality (bio15) of 5 to 35 mm, precipitation of wettest quarter (bio16) of 250 to 700 mm, and precipitation of driest quarter (bio17) of 110 to 300 mm.

### 3.3. Current Potential Distribution

The 10th percentile training presence logistic threshold was 0.1893. According to this, the distribution of *A. custos* was reclassified into four classes (Figure 5). Most of the highly suitable areas (0.6–1) were located in Hebei, Shandong, Anhui, Jiangsu, and parts of in Henan, Hubei, Guizhou, Zhejiang, and Taiwan Provinces. The moderately suitable regions (0.4–0.6) included most of the eastern and central areas (Liaoning, Shanxi, Shaanxi, Hubei, eastern Sichuan, Guizhou, Hunan, Jiangxi, and Fujian Provinces), and parts of Taiwan Province. Marginally suitable areas (0.1893–0.4) were regions of northern China, such as Xinjiang, Inner Mongolia, Heilongjiang, Jilin Provinces, most of southern China (Yunnan, Guangxi, Guangdong, Taiwan Provinces), and a few places in southeastern Tibet. The unsuitable habitats of *A. custos* (<0.1893) were mostly in western China and some in the far north of China. The area of potentially highly suitable areas was approximately 1.10 × 10^6^ km^2^, accounting for about 11.46% of the total land area of China. The area of moderate and marginal suitability habitats were approximately 2.50 × 10^6^ km^2^ each, accounting for about 26.04% of the total area each. Overall, the area of suitable habitats for *A. custos* under the current climate accounts for about 63.54% of the total land area of China (Table 2).

### 3.4. Future Potential Distribution

The potential distribution of *A. custos* under future climate scenarios is significantly different from that under the current climatic conditions (Figure 6). In general, (i) the suitable areas under current climate may massively decrease and highly suitable regions may vanish; (ii) some unsuitable regions in northern China may become suitable habitats under future climatic scenarios.

In the 2050s, most of the moderately suitable regions will be in northeastern Inner Mongolia and northwestern Heilongjiang, and with smaller areas in Jilin and Xinjiang Provinces. Marginally suitable areas will be in Xinjiang, Tibet, Qinghai, Sichuan, and Gansu Provinces. With the range from RCP2.6 to RCP8.5, the suitable regions would be smaller. The area of moderately suitable habitats for *A. custos* under RCP2.6 is approximately 1.80 × 10^5^ km^2^, accounting for about 1.88% of the total land area of China. Compared to the area of moderately suitable habitats under the current climatic condition, this area is reduced by 24.16%.

In the 2070s, the distribution of *A. custos* will be approximately the same as that in the 2050s. Under RCP2.6, the moderately suitable area will be the biggest, which is about 1.99 × 10^5^ km^2^, accounting for about 2.08% of the total land area. Compared to the area of moderately suitable habitats in the 2050s, the area would have increased by 0.2%. The general area of suitable regions in the 2070s would be less than that of the 2050s, excluding that under RCP2.6.

## 4. Discussion

The current study firstly focuses on the potentially suitable areas of *A. custos* in China. We applied MaxEnt to predict the potential distribution area of *A. custos* based on occurrence data and environmental variables under current and future climatic conditions in China. The evaluation result of model proves the high performance of the current model, indicated by that predictive AUC ratio is significantly greater than the random ratio.

The annual mean temperature (bio1) and the annual precipitation (bio12) are the main constraining factors for the distribution of *A. custos*. The survival rate of *A. custos* is highest in regions with medium precipitation (800 mm–1000 mm), which include most of the potentially highly suitable regions both under current and future climate conditions (current: Hebei, Shandong, Henan Provinces; future: Heilongjiang, central Sichuan, southern Qinghai, and Gansu Provinces). Therefore, areas with medium precipitation should be focused on in the future. In this study, the optimal range of annual mean temperature is between 7.5 to 15 °C. Zhou et al. [69] set three temperature gradients (20 °C, 25 °C, and 30 °C) to study their effects on *A. custos.* The results suggested that adults had the longest lifespan and highest survival rate at 20 °C. In addition, Liao and his colleague [70] studied on the growth and development of *A. custos* at temperature conditions of 10 °C and 15 °C. They found that *A. custos* can develop normally at both temperatures, but developmental duration is significantly prolonged with decreasing temperature. These results are nearly consistent with the predicted optimal temperature of *A. custos* in the current model, which has proved that our results are credible.

Qin et al. [71] predicted the potential geographical distribution of FAW in China under the current climatic condition based on its year-round and seasonal distribution. Most medium and low suitable areas of FAW nearly overlap with high suitability areas of *A. custos* (Guizhou, Hubei, Anhui, Jiangsu, Henan, and Shandong Provinces). Highly suitable areas for FAW mostly overlap with *A. custos*’s medium and low suitability habitats (Yunnan, Guangxi, Guangdong, Fujian, Zhejiang, Jiangxi, and Hunan Provinces). Therefore, under the current climatic conditions, releasing *A. custos* in Fujian, Zhejiang, Jiangxi, Hunan, and the southeast of Sichuan Provinces could be useful. However, IPCC estimated that the average global temperature will rise by at most 2.6–4.8 °C and at least 0.3–1.7 °C in the 21st century, which will trigger radically different patterns of distribution [72]. With rise in temperature, the temperature of currently highly suitable habitats may not remain suitable for *A. custos* survival. Our results show that most of the current highly suitable areas will decrease and mainly shift towards northeastern China under climate change scenarios of RCP2.6, 4.5, 8.5 in the 2050s and 2070s. However, these high latitude areas are not suitable for FAW survival, as reported by Xie et al. [73]. Therefore, there is no need to apply *A. custos* for bio-control in the northeast of China under future climatic scenarios.

Although this model has high credibility, there are still some deficiencies. Generally, ecological niche model is mainly affected by abiotic and biotic factors [74], whereas biotic interactions are often excluded from ENMs as they are difficult to quantify [75]. We only considered climatic and topographical variables in current study. Actually, there are many other factors also influencing the result of model, such as land use [76], edaphic variables [77], and vegetation [43]. In addition, the calibration area that has been accessible to the species during its evolutionary history will impact the final results of the model prediction either [78]. However, in our study, we did not compare calibration area of different sizes, so perhaps the current choice is not optimal.

In general, ENMs are effective prediction tools to grasp the potential distribution area of species, and are widely utilized in the field of ecology. In order to improve model accuracy, we suggest various factors that affect the ability of species distribution should be considered. Moreover, as an important predatory insect in agriculture and forestry, *A. custos* needs to be paid more attention in the future.

## 5. Conclusions

This study on the potentially suitable habitats of *A. custos* in China indicates that the annual mean temperature and annual precipitation are the most important environmental factors that influence the distribution of *A. custos.* The highly suitable regions for *A. custos* are in Hebei, Shandong, Anhui, Jiangsu, Henan, Zhejiang, and Guizhou Provinces, under the current climatic conditions. Under climate change, the originally suitable areas will decrease, but some currently unsuitable regions (Heilongjiang, Inner Mongolia) may become suitable. Considering the suitable distribution area of FAW, *A. custos* is suggested to be released in Fujian, Zhejiang, Jiangxi, Hunan, and the southeast of Sichuan Provinces under the current climatic condition. Under future climatic scenarios, it may not be necessary to apply *A. custos* to control FAW in the high latitude areas. 

## Figures and Tables

**Figure 1 insects-11-00674-f001:**
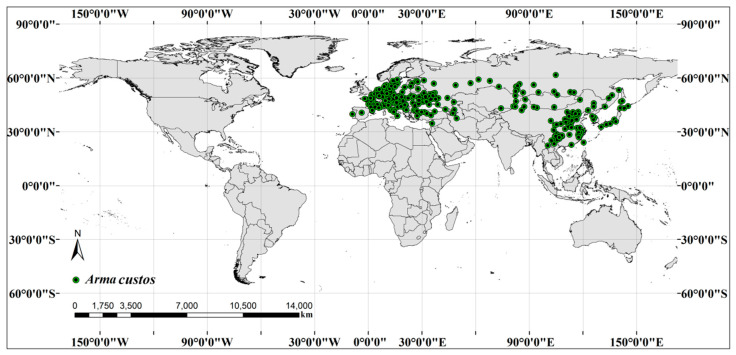
Worldwide geographic distribution records of *A. custos*.

**Figure 2 insects-11-00674-f002:**
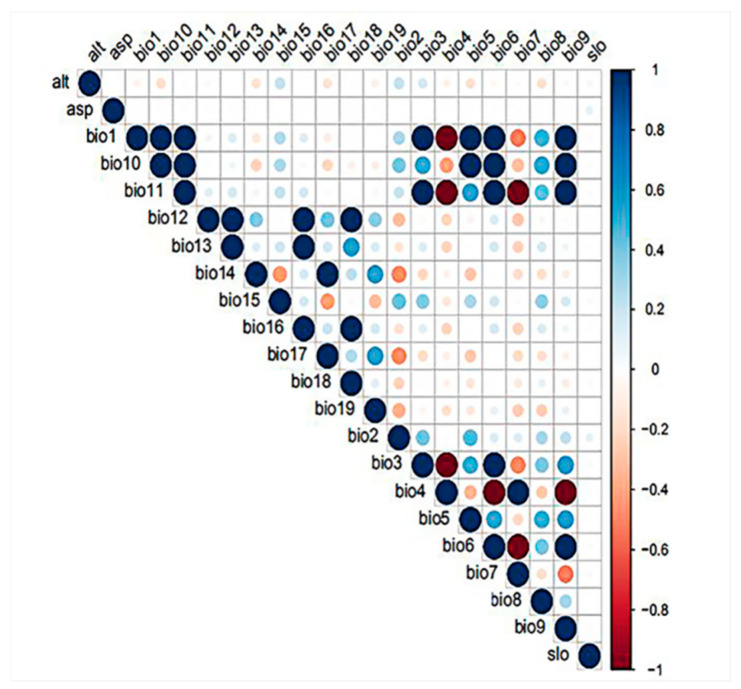
Pearson correlation analysis for environmental variables. The vertical scale represents the value of the correlation coefficients, and the absolute values represent the magnitude of correlation. The pair of high correlated variables (│r│ ≥ 0.8) is represented by a circle that is magnified by 1.5 times and its color is darker than others.

**Figure 3 insects-11-00674-f003:**
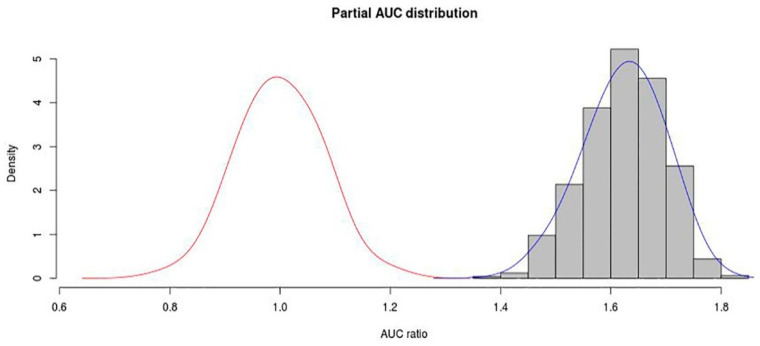
The result of pAUC for *A. custos*. The shaded bars with bell-shaped curve shows the frequency distribution of the ratios between AUC from model prediction and AUC random, while the bell-shape curve on the left side represents the AUC ratios for random models.

**Figure 4 insects-11-00674-f004:**
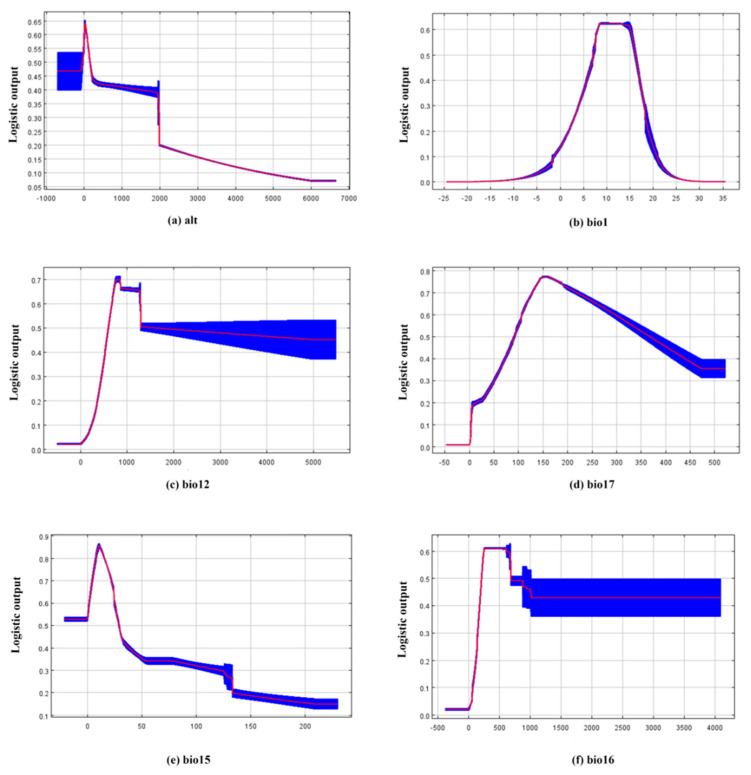
Response curves reflecting the relationship between the potential distribution of *A. custos* and environmental variables. The curves show the mean response of the 10 replicate MaxEnt runs (red) and the mean±SD (blue). (**a**) Altitude; (**b**) The annual mean temperature; (**c**) The annual mean precipitation; (**d**) Precipitation seasonality; (**e**) Precipitation of wettest quarter; (**f**) Precipitation of driest quarter.

**Figure 5 insects-11-00674-f005:**
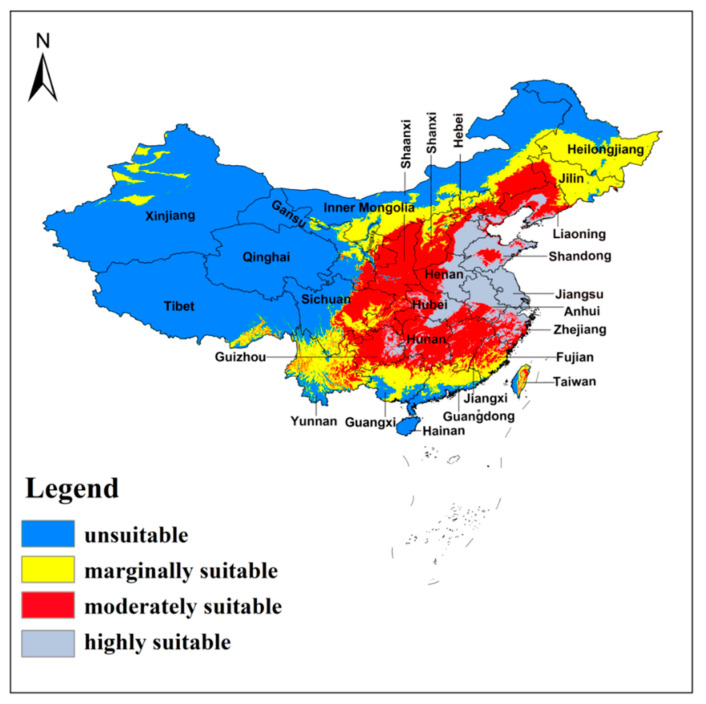
The potential distribution of *A. custos* under current climate in China. The shade of color represents the level of suitability.

**Figure 6 insects-11-00674-f006:**
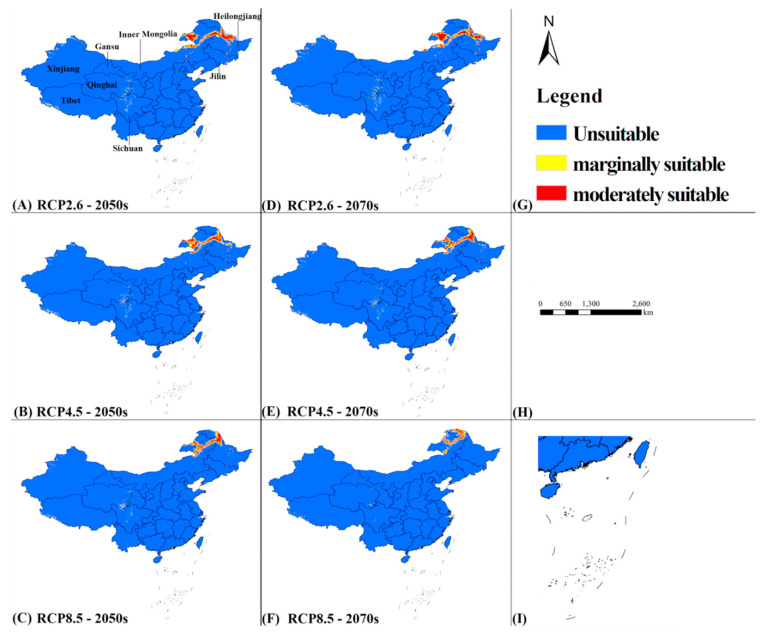
The potential distribution of *A. custos* under future scenarios in China. Pictures from (**A**–**F**) represent the potential distribution of *A. custos* in the 2050s and 2070s under three climate scenarios; (**G**) The legend; (**H**) Scale bar; (**I**) South China Islands.

**Table 1 insects-11-00674-t001:** The contributions of environmental variables.

Variable	Percent Contribution (%)
Annual Mean Temperature (bio1)	58.4
Annual Precipitation (bio12)	24.8
Precipitation Seasonality (bio15)	1.9
Precipitation of Wettest Quarter (bio16)	4
Precipitation of Driest Quarter (bio17)	8.6
Altitude (alt)	2.3

**Table 2 insects-11-00674-t002:** Area of habitats with different suitability for *A. custos* under current and future climatic scenarios (km^2^). The values in brackets represent the proportion of the corresponding area to the total area.

Suitability Grade	Current Climate	RCP2.6 2050s	RCP4.5 2050s	RCP8.5 2050s	RCP2.6 2070s	RCP4.5 2070s	RCP8.5 2070s
None	~3.50 × 10^6^ (36.46%)	~9.27 × 10^6^ (96.66%)	~9.33 × 10^6^ (97.19%)	~9.39 × 10^6^ (97.78%)	~9.24 × 10^6^ (96.23%)	~9.34 × 10^6^ (97.30%)	9.43 × 10^6^ (98.21%)
Marginal	~2.50 × 10^6^ (26.04%)	~1.40 × 10^5^ (1.46%)	~1.20 × 10^5^ (1.25%)	~8.84 × 10^4^ (0.92%)	~1.62 × 10^5^ (1.69%)	~1.24 × 10^5^ (1.29%)	~7.73 × 10^4^ (0.81%)
Moderate	~2.50 × 10^6^ (26.04%)	~1.80 × 10^5^ (1.88%)	~1.50 × 10^5^ (1.56%)	~1.25 × 10^5^ (1.30%)	~1.99 × 10^5^ (2.08%)	~1.35 × 10^5^ (1.41%)	~9.41 × 10^4^ (0.98%)
High	~1.10 × 10^6^ (11.46%)	0	0	0	0	0	0

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
