# Peer review of "Identifying Potentially Climatic Suitability Areas for Arma custos (Hemiptera: Pentatomidae) in China under Climate Change"

_insects, 2020, doi:10.3390/insects11100674_

Round 1
Reviewer 1 Report
I am satisfied by the revisions provided by the authors. I believe they improved the interpretation of their results and provide relevant method and outcome to evaluate the relevance of using insect species as biocontrol.
I would recommend a careful proof reading looking for typos. Here are some typos that I found during my reading but I may have missed others.
line 239: "Person" must be "Pearson"
line 415: remove "in"
Reviewer 2 Report
The revised version is done well.
I have some minor corrections made on the file.
It would be better, if the manuscript can be polished by a professional English editor.
